# Computational Investigation of Smooth Muscle Cell Plasticity in Atherosclerosis and Vascular Calcification: Insights from Differential Gene Expression Analysis of Microarray Data

**DOI:** 10.3390/bioengineering12111223

**Published:** 2025-11-09

**Authors:** Daniel Liu, Jimmy Kuo, Chorng-Horng Lin

**Affiliations:** 1Department of Biomedical Sciences, Da-Yeh University, 168 University Road, Dacun, Changhua 51591, Taiwan; danielyt@mail.dyu.edu.tw; 2Department of Planning and Research, National Museum of Marine Biology and Aquarium, Pingtung 94450, Taiwan; jimmy@nmmba.gov.tw; 3Graduate Institute of Marine Biology, National Dong Hwa University, Pingtung 94450, Taiwan

**Keywords:** smooth muscle cells, microarray, differentiation, dedifferentiation, differentially expressed genes, random forests

## Abstract

The dedifferentiation of smooth muscle cells (SMCs) is the main cause of atherosclerosis and vascular calcification. This study integrated the gene expression data of multiple microarrays to identify relevant marker molecules. A total of 72 Gene Expression Omnibus (GEO) samples (GSM) were collected from 10 gene expression data series (GSE) and divided into five groups: non-SMC, SMC, atherosclerotic SMC (SMC-ath), calcified SMC (SMC-calc), and treated SMC (SMC-t). The SMC-t group included synthetic SMCs that had undergone treatment to inhibit proliferation, migration, or inflammation. The gene expression data were merged, normalized, and batch effects were removed before differential gene expression (DGE) analysis was performed via linear models for microarray data (limma) and statistical analysis of metagenomic profiles (STAMPs). The genes with expressions that significantly differed were subsequently subjected to protein-protein interaction (PPI) and functional prediction analyses. In addition, the random forest method was used for classification. Twelve proteins that may be marker molecules for SMC differentiation and dedifferentiation were identified, namely, Proprotein convertase subtilisin/kexin type 1 (PCSK1), Transforming growth factor beta-induced (TGFBI), Complement C1s (C1S), Phosphomannomutase 1 (PMM1), Claudin 7 (CLDN7), Calcium binding and coiled-coil domain 2 (CALCOCO2), SAC3 domain-containing protein 1 (SAC3D1), Natriuretic peptide B (NPPB), Monoamine oxidase A (MAOA), Regulator of the Cell Cycle (RGCC), Alpha-crystallin B Chain (CRYAB), and Alcohol dehydrogenase 1B (ADH1B). Finally, their possible roles in SMCs are discussed. This study highlights the feasibility of bioinformatics analysis for studying SMC dedifferentiation.

## 1. Introduction

SMCs cultured after several passages are likely to dedifferentiate from the contractile phenotype to the synthetic phenotype, during which the number of myofilaments decreases and the number of organelles in the cells increases upon proliferation [1]. The dedifferentiation process of SMCs, which may lead to the development of foam cells, macrophages, myofibroblasts, osteoblasts, senescent cells, and mesenchymal cells, plays major roles in vascular remodeling and repair during development and injury [2,3,4,5]. However, smooth muscle dedifferentiation is often the cause of vascular-related diseases, such as the accumulation of lipids in the vascular endothelium in atherosclerosis, the narrowing of vascular channels caused by inflammation and fibrosis, and the accumulation of calcium phosphate in the intima and media of blood vessels due to vascular calcification, causing arterial stiffness [3,4,6,7,8,9].

SMCs located in blood vessels, bronchi, and visceral organs show tissue-specific differences in gene expression patterns and might vary in their response to vascular injury [10]. Cholesterol can induce the dedifferentiation of SMCs, and monocyte chemotactic protein-1-induced protein 1 (MCPIP1) plays an important role in this dedifferentiation process, ultimately leading to atherosclerosis [11]. In addition, intracranial aneurysms are the main cause of subarachnoid hemorrhage, and the dedifferentiation of SMCs has become the main histopathological feature of intracranial aneurysms. Platelet-derived growth factor (PDGF-BB) is expressed in vascular endothelial cells, where it promotes the dedifferentiation and proliferation of smooth muscle cells and induces the expression of proinflammatory genes in these cells in vitro [12]. Retinoic acid (RA) signaling is a regulator of SMC differentiation and is dysregulated in human atherosclerosis [13]. Furthermore, NOTCH signaling inhibition is responsible for SMC dedifferentiation into macrophage-like cells [14]. The identification of genes related to SMC differentiation could provide insights into treatment strategies for SMC-related vascular diseases. The gene expression data from microarray analysis in the GEO database [15] provide considerable gene expression information. However, integrated SMC dedifferentiation analysis using multiple microarray datasets is still in its early stages.

In this study, we collected 72 GSM samples from 10 GEO datasets. The expression data were downloaded, merged, normalized, and grouped, followed by DGE, protein–protein interaction (PPI), and functional analyses. We also applied a random forest algorithm for gene selection and classification [16]. In this study, we applied multiple microarray datasets to investigate the plasticity of SMCs in atherosclerosis and vascular calcification.

## 2. Methods

### 2.1. Retrieval and Processing of GEO Microarray Datasets

We downloaded the gene expression datasets from the GEO database [15] with the GEOquerry package [17,18] in R 4.4.1 [19] on the basis of the following keywords: human, smooth muscle cell, atherosclerosis, and calcification. The inner_join command was applied for merging data, with two sets of data each time. After it was integrated into the data, normalization and batch effect removal were performed, and the commands normalizeBetweenArrays with the quantile method and ComBat were applied for analysis. A total of 72 GSM samples from 10 GSE datasets [20,21,22,23,24,25,26,27,28,29] were selected and grouped into five groups for further analysis: non-SMC, SMC, atherosclerotic SMC (SMC-ath), calcified SMC (SMC-calc), and synthetic SMC. These samples were treated by inhibiting proliferation, migration, or inflammation (SMC-t) (Table 1 and Figure 1).

### 2.2. Differentially Expressed Genes (DEGs) and PPI Analyses

The limma R package was applied for DGE of pairwise comparisons [30], and Benjamini-Hochberg (BH) false discovery rate (FDR) [31] correction was applied with adjusted *p* values < 0.05 considered significant. The log2-fold changes (logFC, log2 transformation) indicate the relative fold change in gene expression. The difference between the means of gene expression was pairwise compared via STAMP [32] with Welch’s t-test, followed by BH FDR correction (q < 0.05, the false-positive rate).

String 12.0 [33] and Cytoscape 3.10.3 [34] were employed for the PPIs of the top 50 DEGs and the 12 candidate genes. All genes were input simultaneously to retrieve known and predicted interactions under a Homo sapiens (human) setting. The minimum required interaction score (confidence threshold) was set to “medium confidence 0.4” to balance sensitivity and specificity. The Markov cluster algorithm (MCL) was further applied to identify functional protein clusters based on interaction networks at an inflation parameter of 3 [35]. For PCSK1, TGFBI, C1S, SAC3D1, and RGCC genes, they did not form direct connections to other seed genes in the global PPI. Therefore, separate subnetworks were generated for each by inputting the gene individually and extracting their top 10 interactors. All visualizations were generated using STRING’s built-in network inspection function.

### 2.3. Reactome Pathway Enrichment Analysis

The clusterProfiler [36] and ReactomePA [37] R packages were used for reactome pathway analysis [38]. The difference between the means of the reactome pathway was first mapped to the gene expression table via Entrez id and then pairwise compared via STAMP with Welch’s t-test, followed by BH FDR correction (q < 0.05, the false-positive rate).

### 2.4. Random Forest Analysis

The R packages randomForest 4.7-1.2 [39] and caret 7.0-1 [40] were applied for random forest [16] analysis, a machine learning algorithm with training and randomForest functions to classify DEGs in SMC dedifferentiation.

## 3. Results

### 3.1. Strategy

Figure 1 shows the workflow of the study. A total of 72 gene expression samples from 10 GSE datasets (Table 1) were collected, merged based on probe id, gene symbol or Entrez id, normalized, batch effect removed, and grouped. Principal component analysis (PCA) was performed to check the batch effect removal before and after. After batch correction, batch-driven separation is eliminated, and the principal variation among samples is now associated with biological grouping. However, it is likely that the biological groupings remain partially resolved due to biological heterogeneity. For the merge step, we used the inner_join command to merge columns (samples) with matching rows (probe id, gene symbol or Entrez id). Because different platforms are used for microarray assays, the probe ID and gene number are varied and not completely matched. Finally, 11,195 genes (Entrez id) were obtained, and the table of gene expression information was applied for further DGE analysis by limma and STAMP to perform pairwise comparisons. The functional prediction of PPIs and reactome pathways was investigated based on the DEGs. The expression data of 11,195 genes were subjected to random forest classification to identify key genes distinguishing the groups. Finally, the top two genes identified by each method in the SMC and SMC-clac groups were chosen, providing 12 potential markers of SMC dedifferentiation.

### 3.2. Identification of DEGs and PPI Network Analysis

DEGs were analyzed via both the limma and STAMP programs. Statistically significant DEGs were identified in only two pairwise comparisons: SMC vs. non-SMC and SMC-calc vs. SMC. Figure 2 displays the top 50 DEGs from these analyses. Among these DEGs, 21 DEGs were common to the SMC vs. non-SMC comparison, whereas 34 DEGs overlapped in the SMC-calc vs. SMC comparison, as identified by both limma and STAMP. In the SMC vs. non-SMC comparison, the top two upregulated genes were TGFBI and NPPB according to limma and TGFBI and CRYAB according to STAMP. For the SMC-calc vs. SMC comparison, the top two upregulated genes were MAOA and RGCC, which were identified based on limma, and RGCC and ADH1B, which were identified based on STAMP. Both the TGFBI and RGCC genes were identified via the limma and STAMP tools.

The PPI network of the top DEGs was analyzed via STRING and visualized via Cytoscape (Figure 3). To compare the changes in upregulated proteins located in muscle tissue versus the extracellular compartment, we defined a score greater than 2.0 as indicating a protein likely present in these locations. We then calculated the ratio of upregulated proteins. The proportion of upregulated proteins in muscle tissue decreased from 70% (SMC vs. non-SMC) to 40% (SMC-calc vs. SMC) based on limma analysis and from 60% to 42% based on STAMP analysis. The proportion of upregulated proteins in the extracellular compartment changed from 48% (SMC vs. non-SMC) to 42% (SMC-calc vs. SMC) according to limma, whereas STAMP analysis revealed no change, remaining at 32%. These results suggest that SMCs undergoing calcification shift toward a noncontractile phenotype. Furthermore, the MCL clustering algorithm was used to identify functional clusters. The results revealed clusters associated with aortic aneurysm and elastic fiber formation (LOXL1, FBN1, CD248, TIMP2, MFAP5, SMAD3, SERPINE2, TGFBI, and COPZ2) and DNA methylation (MAPRE3, H2BC21, DNMT3B, and ASPH) in the SMC vs. non-SMC comparison based on limma. In the SMC-calc vs. SMC comparison based on limma, clusters related to high-density lipoprotein (HDL) assembly (PF4V1, APOA1, LPXN, AFP, A2M, and MMP1) and amyloidosis (CX3CL1, APOE, PLIN3, TGFBI, and CH25H) were identified.

### 3.3. Reactome Pathway

The results of the reactome pathway enrichment analysis are presented in Figure 4. In the comparison between the SMC and non-SMC groups, the most significantly enriched pathways identified via the limma approach included rRNA processing, rRNA processing in the nucleus and cytosol, and the major pathway of rRNA processing in these compartments. In contrast, analysis with STAMP highlighted immune system processes, posttranslational protein modifications, and signal transduction as the most significant pathways. When comparing SMC-calc to SMC, limma identified extracellular matrix organization, extracellular matrix degradation, and response to elevated platelet cytosolic Ca^2+^ as the most enriched pathways, whereas STAMP highlighted metabolism, lipid metabolism, and disease-related pathways. The key distinction between the two methods lies in their analytical approach: limma enrichment was based on significantly upregulated DEGs, whereas STAMP first transformed the original expression dataset into a reactome pathway table before performing pairwise comparisons. Notably, limma-based pathway enrichment strongly suggested that SMCs in the SMC-calc group exhibited characteristics of a synthetic phenotype, which is associated with vascular disease progression.

### 3.4. Random Forest Algorithm

We employed the random forest algorithm to characterize DEGs for group classification. The analysis was conducted via the train and randomForest functions, with overall statistics and class-specific results generated via the confusionMatrix function (Appendix A). The parameter mtry (the number of variables randomly sampled as candidates at each split) was set to 11,195 for the training function and 12 for the randomForest function (Appendix A). Since the training function utilized the entire dataset for training, it achieved an accuracy of 1 (Appendix A), with a 100% correct prediction rate. In contrast, the randomForest function yielded an accuracy of 0.6842, with a statistically significant *p* value of 0.01874 (*p* < 0.05) (Appendix A). Using the training function, PCSK1 ranked as the top gene across all five groups, whereas TGFBI and C1S were the second-highest ranking genes in the SMC and SMC-calc groups, respectively (Table 2 and Table 3), suggesting the contribution of these genes in predictive classification. With the randomForest function, the top two genes identified for the SMC group were PMM1 and CLDN7 (Table 4), whereas CALCOCO2 and SAC3D1 were the top genes for the SMC-calc group (Table 5). Again, higher gene scores suggest that the genes strongly predict a class.

### 3.5. Candidate Genes

Finally, we generated a boxplot for the expression of the 12 marker genes mentioned previously (Figure 5). These genes are PCSK1, TGFBI, PMM1, CLDN7, NPPB, and CRYAB in SMC and PCSK1, C1S, CALCOCO2, SAC3D1, MAOA, RGCC, and ADH1B in SMC-calc. The TGFBI gene was the only gene selected by the random forest, limma and STAMP programs for the SMC group.

Protein–protein interaction (PPI) network analysis of the 12 candidate genes, using STRING and MCL clustering, revealed six functional clusters separated by dashed lines; the thickness of the lines reflects the confidence level of the interaction (Figure 6). PCSK1, RGCC, SAC3D1, TGFBI, and C1S genes did not form highly connected modules in the global network. Therefore, individual analyses were performed on these genes as peripheral modules (Figure 6). The dominant ADH1 and MAOA cluster is predominantly composed of members of the alcohol and aldehyde dehydrogenase family and other oxidoreductases, characterized as being involved in drug metabolism. In addition, the CLDN7 cluster is associated with the positive regulation of blood–brain barrier permeability, the AOC1 cluster with amine metabolism and local vascular regulation, the NPPB cluster with natriuretic peptide as a vasoactive factor, the CALCOCO2 cluster with autophagy and immunity, and the CRYAB cluster with eye lens protein function. Peripheral modules identified by MCL included clusters involved in peptide hormone regulation (centered on PCSK1), phosphorylation of Early Mitotic Inhibitor 1 (Emi1) (RGCC cluster), extracellular matrix remodeling (TGFBI cluster), transcription export complex 2 related to ribosomal and RNA processing (SAC3D1 cluster), and complement-mediated immune response (C1S cluster). Overall, this network analysis shows that these genes participate in coordinated yet diverse biological modules, encompassing processes such as vascular remodeling, metabolic regulation, cellular homeostasis, and immune defense. This suggests their functional connectivity in the mechanisms of vascular disease and their potential value as candidate genes for translational biomarkers.

## 4. Discussion

The present study integrates multiple microarray datasets to identify key marker molecules associated with smooth muscle cell (SMC) dedifferentiation: a fundamental process in atherosclerosis and vascular calcification. By integrating multiple microarray datasets, we identified 12 potential marker genes associated with different stages of SMC dedifferentiation. Our findings provide valuable insights into the molecular mechanisms driving this phenotypic transition and highlight potential therapeutic targets. For example, the downregulation of muscle genes (STRING) and reactome pathways of extracellular matrix organization, extracellular matrix degradation, and response to elevated platelet cytosolic Ca^2+^ are consistent with previous reports highlighting the phenotypic switch of SMCs from a contractile to a synthetic state in vascular disease [41,42].

Among the 12 identified markers, several have been previously implicated in SMC dedifferentiation and vascular pathology. PCSK1, a proprotein convertase, has been identified as a promising therapeutic target for cardiovascular diseases [43]. TGFBI is an extracellular matrix protein that increases the proliferation of airway SMCs [44], and it has been associated with stimulating SMC migration [45]. PMM1 is an enzyme involved in the synthesis of GDP-mannose, and plasma mannose levels are related to incident type 2 diabetes and cardiovascular disease [46]. Furthermore, CLDN7, a tight junction protein, suppresses breast cancer invasion and metastasis through the repression of the smooth muscle actin program [47]. NPPB, a hormone, may play a protective role in vascular disease by counteracting the effects of vasoconstrictors [48]. CRYAB is a small heat shock protein that acts as a molecular chaperone, and it is upregulated in injury-induced vascular SMC proliferation [49]. Additionally, complement activation contributes to vascular damage in atherosclerosis, and C1S may be involved in promoting inflammation and SMC dysfunction in atherosclerotic lesions [50]. CALCOCO2, also known as NDP52, has been studied primarily in the context of autophagy and innate immunity [51]. However, its role in SMCs is not well defined. SAC3D1 is involved in the cell cycle, centrosome duplication and spindle formation [52]. Although its role in SMCs is uncertain, it has been used as a prognostic marker in cancer [53]. MAOA is an enzyme that breaks down neurotransmitters such as serotonin, dopamine, and norepinephrine. MAOA affects the contractile activity of smooth muscle gastric tissues through 5-hydroxytryptamine [54]. RGCC may promote injury-induced vascular neointima formation by mediating SMC proliferation and migration, thus facilitating atherosclerosis development [55]. The role of ADH1B in SMCs is not well understood, although it is downregulated in airway smooth muscles by epidermal growth factor and interleukin-1β [56].

Among the 12 markers identified in this study, many are already known to play important roles in smooth muscle cell biology and vascular disease, which shows their value for translation. Our PPI network analysis further showed that these genes are grouped into specific functional clusters, not randomly scattered. These modules include vascular remodeling and fibrosis (TGFBI), metabolic and redox functions (ADH1/MAOA cluster), hormone-related pathways (PMM1 and NPPB), cell junction structure (CLDN7), immune and autophagy functions (C1S and CALCOCO2), and stress response (CRYAB). For example, the ADH1 and MAOA cluster is related to metabolism and vascular diseases, and may be useful as a biomarker or treatment target. The CLDN7, NPPB, CALCOCO2, and CRYAB clusters have functions in maintaining cell stability, regulating blood vessel contraction, protecting against immune stress, and helping the cell withstand damage. These network results confirm that the genes work together in connected biological processes that are key to smooth muscle cell changes and vascular disease. Therefore, using PPI network analysis helps us understand how these genes can be applied, particularly for developing diagnostic tests, assessing risk, or designing individualized treatments for vascular disease [44,45,46,47,48,49,50,51].

This study has several limitations. First, the sample size was relatively small, which may limit the statistical power of subgroup analyses. To address this, we used three different methods to perform differential gene expression analysis to ensure the robustness of the findings. The first was limma, the second was STAMP with pairwise comparison, and the third was machine learning-based classification via the random forest method. Second, our analysis was based on microarray data, which may not capture the full complexity of gene expression changes. For example, alternative splicing programs have been used in SMCs [57,58]. Third, our study focused primarily on gene expression analysis, and further functional studies are needed to validate the role of the identified marker genes in SMC dedifferentiation. For example, single-cell RNA sequencing [13,14] and spatial transcriptomics [59] can help validate and refine the gene markers as well as track cell lineage changes during dedifferentiation. 

In conclusion, this study demonstrated the feasibility of the use of bioinformatics approaches for identifying key molecular markers associated with SMC dedifferentiation. These findings underscore the complexity of SMC phenotypic switching and highlight potential targets for mitigating vascular pathologies. Future research should focus on these findings to explore the translational potential of these markers in clinical settings.

## Figures and Tables

**Figure 1 bioengineering-12-01223-f001:**
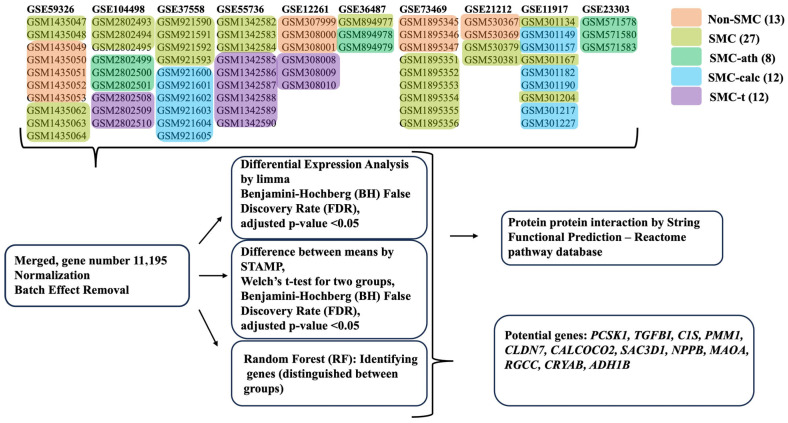
The study design and analysis workflow for gene expression profiling of SMCs across multiple datasets.

**Figure 2 bioengineering-12-01223-f002:**
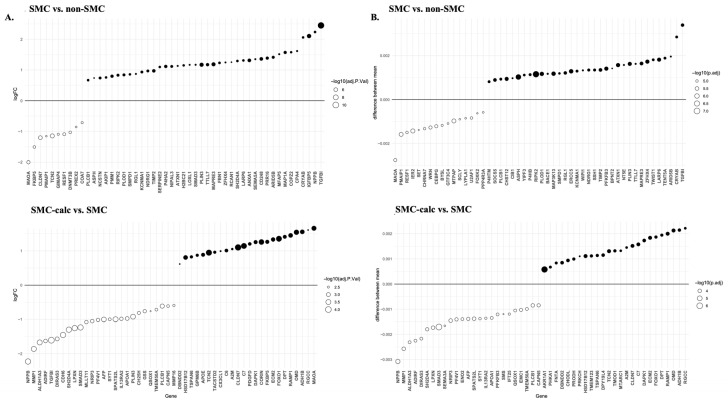
The top 50 DEGs in two pairwise comparisons: SMC vs. non-SMC and SMC-calc vs. SMC. (**A**). The log2-fold change (logFC) of the top 50 DEGs was analyzed by limma. Genes with positive logFCs are upregulated in SMCs, whereas those with negative logFCs are downregulated. The size of each point represents the statistical significance (−log10 adjusted *p* value). (**B**). The difference in the mean expression of the top 50 genes was analyzed via STAMP. The size of each point corresponds to the -log10 adjusted *p* value.

**Figure 3 bioengineering-12-01223-f003:**
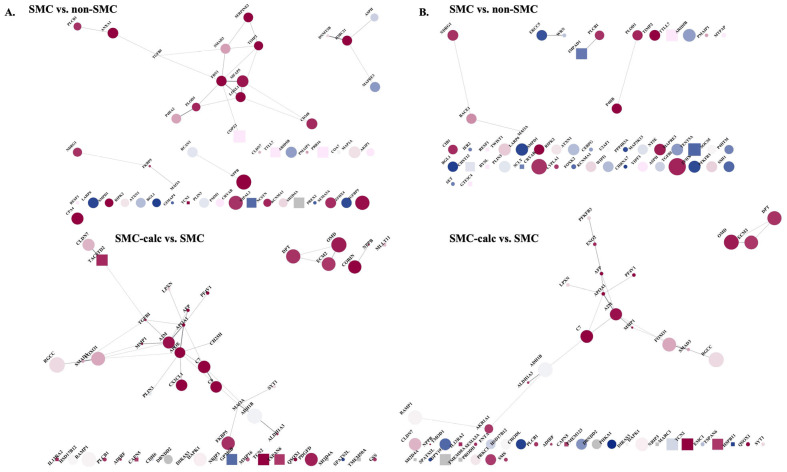
The PPI network of the top 50 DEGs. (**A**). from limma and (**B**). from STAMP in two pairwise comparisons: SMC vs. non-SMC and SMC-calc vs. SMC. The PPI network was analyzed via STRING and visualized via Cytoscape. In the network, the size of each shape corresponds to the gene expression level: larger shapes denote upregulated gene expression (+), whereas smaller shapes indicate downregulated gene expression (−). The circles represent proteins associated with muscle tissue with a score >2.0, and the squares represent proteins with a score <2.0. Additionally, red indicates proteins located in the extracellular compartment with a score >2.0, whereas blue represents those with a score <2.0.

**Figure 4 bioengineering-12-01223-f004:**
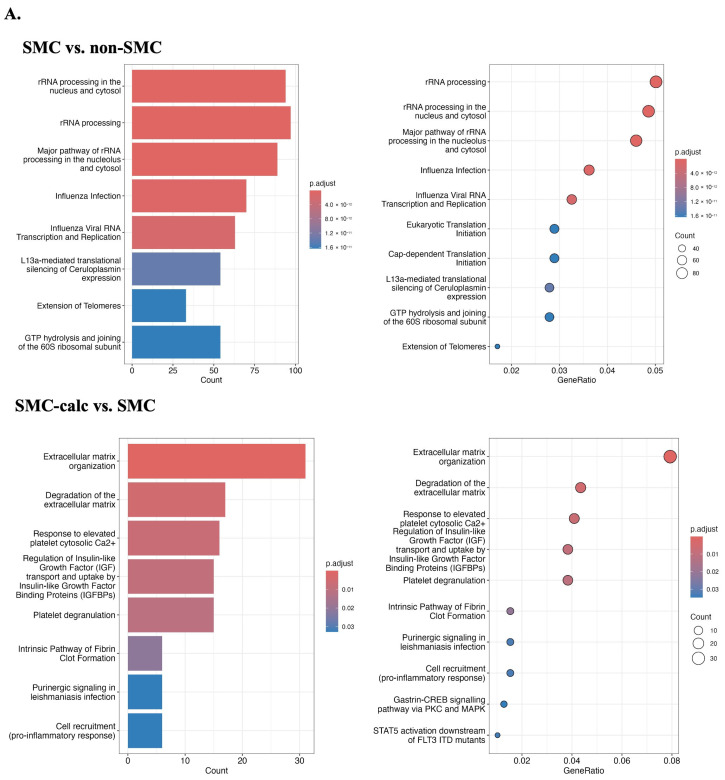
Reactome pathway annotation in two pairwise comparisons: SMC vs. non-SMC and SMC-calc vs. SMC. (**A**). Reactome pathway enrichment analysis of DEGs. The colors represent the statistical significance (adjusted *p* value). (**B**). The reactome pathway was first mapped to the gene expression table via Entrez id and then pairwise compared via STAMP with Welch’s t-test, followed by BH FDR correction (q < 0.05, the false-positive rate). The size of each point corresponds to the -log10 adjusted *p* value.

**Figure 5 bioengineering-12-01223-f005:**
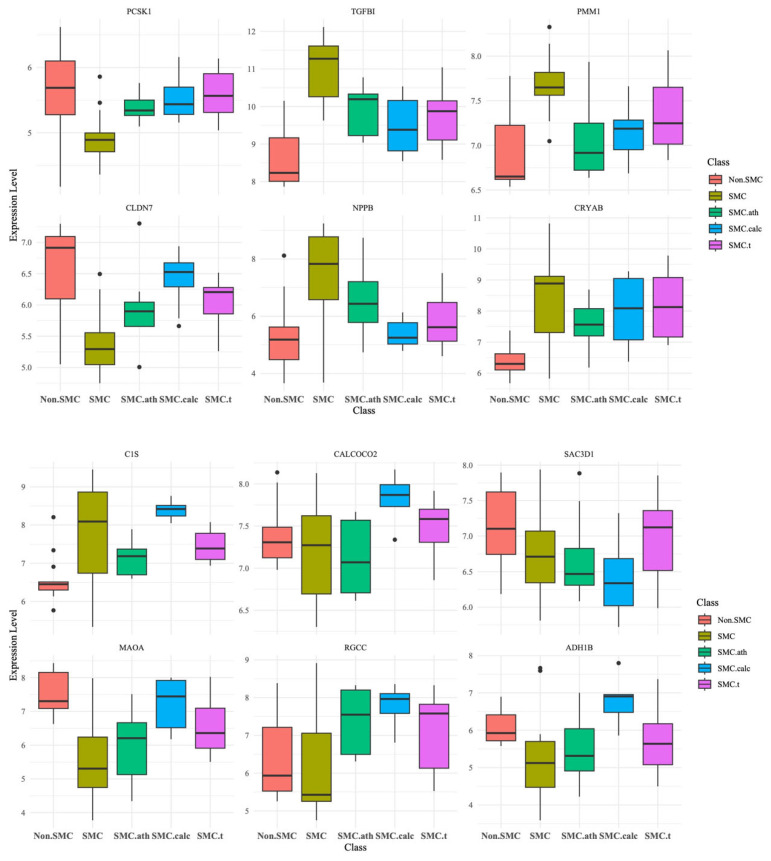
Boxplot of the expression of the 12 marker genes across different groups (non-SMC, SMC, SMC-ath, SMC-calc, and SMC-t). The genes included PCSK1, TGFBI, PMM1, CLDN7, NPPB, and CRYAB in SMC, as well as C1S, CALCOCO2, SAC3D1, MAOA, RGCC, and ADH1B in SMC-calc. PCSK1 is a marker gene found in both SMC and SMC-calc but is shown only once. The *y*-axis represents gene expression levels, whereas the *x*-axis categorizes samples into their respective groups. The different colors indicate different groups, as shown in the legend.

**Figure 6 bioengineering-12-01223-f006:**
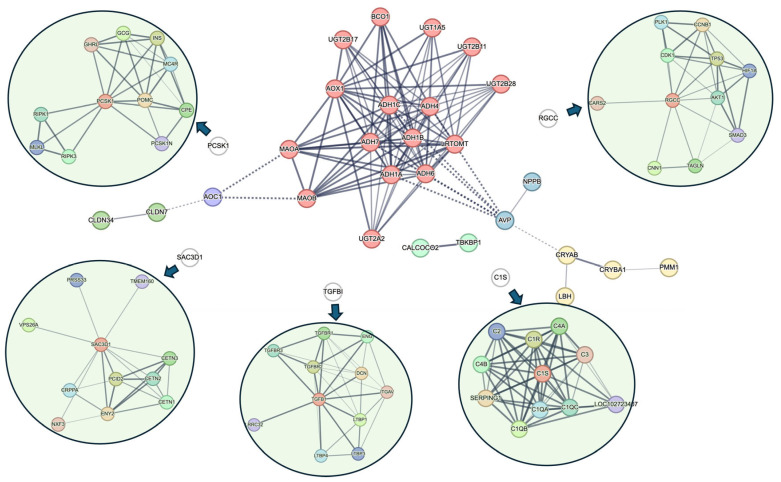
The PPI network of the 12 candidate genes was constructed using the STRING database. The central network represents the result of simultaneously inputting all 12 genes, with the number of visible interactions adjusted and clustering performed using the MCL algorithm. Nodes are colored according to their cluster, with each color representing an independent functional module. The six functional clusters are separated by dashed lines; the thickness of the lines reflects the confidence level of the interactions. Due to the lack of direct STRING interactions with other genes, five genes that could not form clear PPI subnetworks within the main network are displayed separately as peripheral modules. Each subnetwork was constructed based on the top 10 interacting proteins identified from individual STRING analyses. These independent subnetworks are indicated by blue arrows.

**Table 1 bioengineering-12-01223-t001:** Gene expression datasets used in this study and metadata.

GSM	Groups *	GEO	Platform	Genes Nos	References
GSM1435047, GSM1435048, GSM1435062, GSM1435063, GSM1435064	SMC	GSE59326	GPL16025	45033	[26]
GSM1435049, GSM1435050, GSM1435051	non-SMC				
GSM1435052, GSM1435053					
GSM2802493, GSM2802494, GSM2802495	SMC	GSE104498	GPL6884	48803	[22]
GSM2802499, GSM2802500, GSM2802501	SMC-ath				
GSM2802508, GSM2802509, GSM2802510	SMC-t				
GSM921590, GSM921591, GSM921592 GSM921593	SMC	GSE37558	GPL6947	48803	[20]
GSM921600, GSM921601, GSM921602 GSM921603, GSM921604, GSM921605	SMC-calc				
GSM1342582, GSM1342583, GSM1342584	SMC	GSE55736	GPL10558	47231	[24]
GSM1342585, GSM1342586, GSM1342587 GSM1342588, GSM1342589, GSM1342590	SMC-t				
GSM307999, GSM308000, GSM308001	non-SMC	GSE12261	GPL570	54675	[28]
GSM308008, GSM308009, GSM308010	SMC-t				
GSM894977	SMC	GSE36487	GPL96	22283	[25]
GSM894978, GSM894979	SMC-ath				
GSM1895345, GSM1895346, GSM1895347	non-SMC	GSE73469	GPL571	22277	[23]
GSM1895351, GSM1895352, GSM1895353 GSM1895354, GSM1895355, GSM1895356	SMC				
GSM530367, GSM530369	non-SMC	GSE21212	GPL3921	22277	[21]
GSM530379, GSM530381	SMC				
GSM301134, GSM301167, GSM301204	SMC	GSE11917	GPL570	54675	[29]
GSM301149, GSM301157, GSM301182 GSM301190, GSM301217, GSM301227	SMC-calc				
GSM571578, GSM571580, GSM571583	SMC-ath	GSE23303	GPL4372	39096	[27]

* There are five groups: non-SMC, SMC, atherosclerotic SMC (SMC-ath), calcified SMC (SMC-calc), and treated SMC (SMC-t). The SMC-t group represents synthetic SMC that have undergone treatment to inhibit proliferation, migration, or inflammation.

**Table 2 bioengineering-12-01223-t002:** Top 10 features ranked by the training function for classification of the SMC group.

	Non-SMC	SMC	SMC-ath	SMC-calc	SMC-t
PCSK1	62.733	100.000	46.721	78.199	53.004
TGFBI	38.601	43.406	23.182	26.441	31.568
ALDH1A3	31.619	41.766	28.116	33.278	27.762
SH2D4A	29.233	39.417	27.021	23.853	35.339
HSD17B12	29.627	37.103	20.031	34.060	21.032
CLDN7	26.459	35.539	19.963	31.110	0.000
PLIN3	35.008	34.847	24.163	29.595	24.152
CFH	35.554	34.247	10.693	25.307	28.402
RIPK2	31.629	31.768	19.310	25.073	24.740
SMAD3	25.245	31.564	24.974	30.729	13.388

**Table 3 bioengineering-12-01223-t003:** Top 10 features ranked by the training function for classification of the SMC-calc group.

	Non-SMC	SMC	SMC-ath	SMC-calc	SMC-t
PCSK1	62.733	100.000	46.721	78.199	53.004
C1S	42.738	21.013	34.337	51.975	29.658
LUM	34.521	12.726	22.710	46.343	26.576
THY1	25.047	16.462	13.380	40.420	30.583
PPIC	34.105	20.113	21.731	37.905	23.531
DNAAF5	52.480	23.263	28.099	37.518	28.962
NNMT	32.963	25.275	24.937	35.604	21.126
HSD17B12	29.627	37.103	20.031	34.060	21.032
ALDH1A3	31.619	41.766	28.116	33.278	27.762
CLDN7	26.459	35.539	19.963	31.110	0.000

**Table 4 bioengineering-12-01223-t004:** Top 10 features ranked by the randomForest function for classification of the SMC group.

	Non-SMC	SMC	SMC-ath	SMC-calc	SMC-t	MeanDecrease Accuracy	MeanDecrease Gini
PMM1	1.726	2.175	1.001	1.001	−0.640	2.605	0.046
CLDN7	1.989	2.164	0.000	0.000	−1.388	2.104	0.040
CAMK2N1	1.729	1.989	−1.001	0.000	0.000	2.044	0.023
GAMT	0.426	1.986	0.714	1.343	1.001	2.039	0.035
CDH6	0.000	1.967	1.001	1.343	1.001	1.721	0.019
ZNF148	1.388	1.966	0.000	0.000	0.000	1.898	0.023
INHBB	1.314	1.828	−0.905	1.001	1.301	1.801	0.032
PCSK1	1.001	1.801	0.000	1.001	0.000	1.731	0.030
HIP1	0.000	1.734	1.416	−1.001	−1.001	1.796	0.026
HYOU1	1.001	1.731	0.000	0.000	−1.001	1.307	0.017

**Table 5 bioengineering-12-01223-t005:** Top 10 features ranked by the randomForest function for classification of the SMC-calc group.

	Non-SMC	SMC	SMC-ath	SMC-calc	SMC-t	MeanDecrease Accuracy	MeanDecrease Gini
CALCOCO2	−1.001	−0.137	0.000	2.022	0.000	1.590	0.042
SAC3D1	1.001	0.000	1.001	1.688	0.000	1.408	0.016
AARS1	1.001	−1.001	0.000	1.674	−1.001	1.378	0.020
KRT7	1.001	1.685	0.000	1.669	−1.001	1.949	0.018
CYP27A1	−1.001	1.257	0.000	1.667	0.000	1.664	0.017
MRPL34	1.001	0.294	−1.001	1.635	1.001	1.449	0.031
RECQL4	1.214	−0.175	0.000	1.597	0.000	1.408	0.014
RTCB	1.407	0.000	0.000	1.416	0.000	1.614	0.015
YEATS4	1.388	1.001	0.000	1.416	0.000	1.658	0.016
NCAPD2	1.343	1.411	1.001	1.416	0.000	1.414	0.016

## Data Availability

The raw data supporting the conclusions of this article will be made available by the authors on request.

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
