# Peer review of "Computational Investigation of Smooth Muscle Cell Plasticity in Atherosclerosis and Vascular Calcification: Insights from Differential Gene Expression Analysis of Microarray Data"

_bioengineering, 2025, doi:10.3390/bioengineering12111223_

Round 1
Reviewer 1 Report
Comments and Suggestions for Authors
The study merges data from 10 independent GEO datasets (72 samples), which increases statistical power and reduces dataset bias, offering a more robust insight into SMC phenotypic transitions. Integration of DGE (limma + STAMP), PPI, and random forest classification provides a robust and multidimensional exploration of marker genes, gives a better methodological viewpoint. The authors need to address the following in the revision.
- Even after merging, only 72 samples across five groups may lead to underpowered subgroup analyses.
- No wet-lab or in vitro validation (e.g., qPCR, Western blot, or immunostaining) to confirm predicted biomarkers.
- Although batch effects were removed, microarray heterogeneity (different probe designs/platforms) may still confound expression comparisons.
- Extend beyond random forests to explainable deep learning (e.g., SHAP-based interpretability or graph neural networks on PPI networks) to identify causal gene modules and therapeutic targets.
- Future work could validate these markers across single-cell atlases of vascular tissues to pinpoint lineage trajectories during dedifferentiation. Address this.
Author Response
Response to reviewer
Reviewer 1
The study merges data from 10 independent GEO datasets (72 samples), which increases statistical power and reduces dataset bias, offering a more robust insight into SMC phenotypic transitions. Integration of DGE (limma + STAMP), PPI, and random forest classification provides a robust and multidimensional exploration of marker genes, gives a better methodological viewpoint. The authors need to address the following in the revision.
- Even after merging, only 72 samples across five groups may lead to underpowered subgroup analyses.
Response: We thank the reviewer for the comment about the sample size of 72 samples across five groups. We understand that this sample size may reduce the statistical power in subgroup analyses. To reduce this limitation, we used different statistical methods and careful validation to make the results more reliable. However, we agree that using larger sample groups in future studies will be important to confirm and expand these results.
We have made the change with red highlight in the manuscript to address the weakness: (line 377- 382) This study has several limitations. First, the sample size was relatively small, which may limit the statistical power of subgroup analyses. To address this, we used three different methods to perform differential gene expression analysis to ensure robustness of the findings The first was limma, the second was STAMP with pairwise comparison, and the third was machine learning-based classification via the random forest method.
- No wet-lab or in vitro validation (e.g., qPCR, Western blot, or immunostaining) to confirm predicted biomarkers.
Response: We agree with the reviewer that experimental validation (e.g., qPCR, Western blot, immunostaining) is important to confirm predicted biomarkers. While these validation steps were beyond the scope of the current study, we intend to pursue such wet-lab experiments in future work to substantiate the functional relevance of our candidate genes.
To enhance our persuasiveness, we further analyzed the PPIs of these 12 genes. Using string's web pages and databases, and employing functional classification systems such as KEGG, reactome, GO, and pfam, we generated surface-level and even three-dimensional relationship networks to provide insights into the functional systems of these genes. The relevant results and descriptions are listed in point four of the response. Of course, such analysis remains predictive, not direct evidence.
- Although batch effects were removed, microarray heterogeneity (different probe designs/platforms) may still confound expression comparisons.
Response: We acknowledge that while batch effects were addressed using state-of-the-art correction methods, residual confounding due to different microarray platforms and probe designs may still exist. This is an inherent limitation when integrating public datasets generated using heterogeneous platforms. We minimized this as much as possible through rigorous normalization and batch correction pipelines. Principal component analysis (PCA) was performed to check the batch effect removal before and after. Below are the PCA data before and after batch effect removal.
We have made the change with red highlight in the manuscript for the supplementary explanation: (line 140- 145) Figure 1 shows the workflow of the study. A total of 72 gene expression samples from 10 GSE datasets (Table 1) were collected, merged based on probe id, gene symbol or Entrez id, normalized, batch effect removed, and grouped. Principal component analysis (PCA) was performed to check the batch effect removal before and after. After batch correction, batch-driven separation is eliminated and the principal variation among samples is now associated with biological grouping (data not shown). However, it is likely the biological groupings remain partially resolved due to the biological heterogeneity. For the merge step, we used the inner_join command to merge columns (samples) with matching rows (probe id, gene symbol or Entrez id). Because different platforms are used for microarray assays, the probe ID and gene number are varied and not completely matched. Finally, 11195 genes (Entrez id) were obtained, and the table of the gene expression information was applied for further DGE analysis by limma and STAMP to perform pairwise comparisons. The functional prediction of PPIs and reactome pathways was investigated based on the DEGs. The expression data of 11,195 genes were subjected to random forest classification to identify key genes distinguishing the groups.Finally, the top two genes identified by each method in the SMC and SMC-clac groups were chosen, providing 12 potential markers of SMC dedifferentiation.
- Extend beyond random forests to explainable deep learning (e.g., SHAP-based interpretability or graph neural networks on PPI networks) to identify causal gene modules and therapeutic targets.
Response: We thank the reviewer for your valuable suggestion to improve the interpretation of our results. We agree that extending the analysis beyond random forest by using explainable deep learning methods, such as applying graph neural networks (GNNs) to protein-protein interaction (PPI) networks, can give deeper understanding of gene modules and possible therapeutic targets. However, because of the current study scope and the small number of high-confidence marker genes (n=12), direct use of GNN or deep-learning-based module discovery may be limited by the size and density of the network. Therefore, we further analyzed the PPIs of these 12 genes by adding their direct interactors from STRING, KEGG, or reactome databases. This analysis visualizes and annotates the connectivity among the marker genes and their interaction partners in a network context. Although the STRING network shown in this figure is not a graph neural network (GNN), it provides a high-quality biological graph structure that can be used as a foundation for more advanced analyses. In other words, STRING networks contain biological knowledge and functional relationships, and they can be exported and used as input data for machine learning methods, such as GNNs, for node classification, subnetwork discovery, or disease gene prediction. In summary, this functional PPI cluster analysis provides mechanistic hypotheses mapped to biological modules and supports both translational interpretation and future computational modeling, including GNN application if needed.
We have made the change with red highlight in the manuscript:
Figure 6. The PPI network of the 12 candidate genes constructed using the STRING database. The central network represents the result of simultaneously inputting all 12 genes, with the number of visible interactions adjusted and clustering performed using the MCL algorithm. Nodes are colored according to their cluster, with each color representing an independent functional module. The six functional clusters are separated by dashed lines; the thickness of the lines reflects the confidence level of the interactions. Due to the lack of direct STRING interactions with other genes, five genes that could not form clear PPI subnetworks within the main network are displayed separately as peripheral modules. Each subnetwork was constructed based on the top 10 interacting proteins identified from individual STRING analyses. These independent subnetworks are indicated by blue arrows. (Line 311-321)
(Line 106-116) String 12.0 [33] and Cytoscape 3.10.3 [34] were employed for the PPIs of the top 50 DEGs and the 12 candidate genes. All genes were input simultaneously to retrieve known and predicted interactions under a Homo sapiens (human) setting. The minimum required interaction score (confidence threshold) was set to "medium confidence 0.4" to balance sensitivity and specificity. The Markov cluster algorithm (MCL) was further applied to identify functional protein clusters based on interaction networks at an inflation parameter of 3 [35]. For PCSK1, TGFBI, C1S, SAC3D1, and RGCC genes, they did not form direct connections to other seed genes in the global PPI. Therefore, separate subnetworks were generated for each by inputting the gene individually and extracting their top 10 interactors. All visualizations were generated using STRING's built-in network inspection function.
- 12 candidate genes
Finally, we generated a boxplot for the expression of the 12 marker genes mentioned previously (Figure 5). These genes are PCSK1, TGFBI, PMM1, CLDN7, NPPB, and CRYAB in SMC and PCSK1, C1S, CALCOCO2, SAC3D1, MAOA, RGCC, and ADH1B in SMC-calc. The TGFBI gene was the only gene selected by the random forest, limma and STAMP programs for the SMC group.
Protein–protein interaction (PPI) network analysis of the 12 candidate genes, using STRING and MCL clustering, revealed six functional clusters separated by dashed lines; the thickness of the lines reflects the confidence level of the interaction (Figure 6). PCSK1, RGCC, SAC3D1, TGFBI, and C1S genes did not form highly connected modules in the global network. Therefore, individual analyses were performed on these genes as peripheral modules (Figure 6). The dominant ADH1 and MAOA cluster is predominantly composed of members of the alcohol and aldehyde dehydrogenase family and other oxidoreductases, characterized as being involved in drug metabolism. In addition, the CLDN7 cluster is associated with the positive regulation of blood–brain barrier permeability, the AOC1 cluster with amine metabolism and local vascular regulation, the NPPB cluster with natriuretic peptide as a vasoactive factor, the CALCOCO2 cluster with autophagy and immunity, and the CRYAB cluster with eye lens protein function. Peripheral modules identified by MCL included clusters involved in peptide hormone regulation (centered on PCSK1), phosphorylation of Early Mitotic Inhibitor 1 (Emi1) (RGCC cluster), extracellular matrix remodeling (TGFBI cluster), transcription export complex 2 related to ribosomal and RNA processing (SAC3D1 cluster), and complement-mediated immune response (C1S cluster). Overall, this network analysis shows that these genes participate in coordinated yet diverse biological modules, encompassing processes such as vascular remodeling, metabolic regulation, cellular homeostasis, and immune defense. This suggests their functional connectivity in the mechanisms of vascular disease and their potential value as candidate genes for translational biomarkers. (Line 269-297)
- Future work could validate these markers across single-cell atlases of vascular tissues to pinpoint lineage trajectories during dedifferentiation. Address this.
Response: We appreciate the suggestion to validate our markers using single-cell transcriptomic atlases of vascular tissues to map lineage trajectories during dedifferentiation. This would provide valuable cellular resolution and mechanistic insight.
We have made the change with red highlight in the manuscript to address the issue: (line 386- 388) Second, our analysis was based on microarray data, which may not capture the full complexity of gene expression changes. For example, alternative splicing programs have been used in SMCs [57,58]. Third, our study focused primarily on gene expression analysis, and further functional studies are needed to validate the role of the identified marker genes in SMC dedifferentiation. For example, single-cell RNA sequencing [13,14] and spatial transcriptomics [59] can help validate and refine the gene markers as well as track cell lineage changes during dedifferentiation.

Reviewer 2 Report
Comments and Suggestions for Authors
This study uses standard DGE methods to analyze various SMC related datasets. There are thousands of studies analyzing gene expression datasets and these have been published, but the key question is how is this relevant to human health? Meaning how is this translatable to human health?
The authors have made some attempt in the Discussion to discuss the relevance of their findings but they need a clear and thorough paragraph on how their findings are translatable. What is the translational science? How can this help humans in the end? I do not mean to sound harsh, but this type of data mining study is not helpful without some guidelines on how clinicians can use the data to help patients.
Author Response
Reviewer 2
This study uses standard DGE methods to analyze various SMC related datasets. There are thousands of studies analyzing gene expression datasets and these have been published, but the key question is how is this relevant to human health? Meaning how is this translatable to human health?
The authors have made some attempt in the Discussion to discuss the relevance of their findings but they need a clear and thorough paragraph on how their findings are translatable. What is the translational science? How can this help humans in the end? I do not mean to sound harsh, but this type of data mining study is not helpful without some guidelines on how clinicians can use the data to help patients.
Response: We thank the reviewer for this insightful comment regarding the translational relevance of our gene expression findings. we further analyzed the PPIs of these 12 genes by adding their direct interactors from STRING, KEGG, or Reactome databases. This functional PPI cluster analysis provides mechanistic hypotheses mapped to biological modules and supports both translational interpretation and future computational modeling, including GNN application if needed. Further, in Discussion we revise and provide a focused paragraph outlining how our identified SMC marker genes are intimately linked with human disease mechanisms and how they may translate to clinical impact.
We have made the change with red highlight in the manuscript:
Figure 6. The PPI network of the 12 candidate genes constructed using the STRING database. The central network represents the result of simultaneously inputting all 12 genes, with the number of visible interactions adjusted and clustering performed using the MCL algorithm. Nodes are colored according to their cluster, with each color representing an independent functional module. The six functional clusters are separated by dashed lines; the thickness of the lines reflects the confidence level of the interactions. Due to the lack of direct STRING interactions with other genes, five genes that could not form clear PPI subnetworks within the main network are displayed separately as peripheral modules. Each subnetwork was constructed based on the top 10 interacting proteins identified from individual STRING analyses. These independent subnetworks are indicated by blue arrows. (Line 311-321)
(Line 106-116) String 12.0 [33] and Cytoscape 3.10.3 [34] were employed for the PPIs of the top 50 DEGs and the 12 candidate genes. All genes were input simultaneously to retrieve known and predicted interactions under a Homo sapiens (human) setting. The minimum required interaction score (confidence threshold) was set to "medium confidence 0.4" to balance sensitivity and specificity. The Markov cluster algorithm (MCL) was further applied to identify functional protein clusters based on interaction networks at an inflation parameter of 3 [35]. For PCSK1, TGFBI, C1S, SAC3D1, and RGCC genes, they did not form direct connections to other seed genes in the global PPI. Therefore, separate subnetworks were generated for each by inputting the gene individually and extracting their top 10 interactors. All visualizations were generated using STRING's built-in network inspection function.
- 12 candidate genes
Finally, we generated a boxplot for the expression of the 12 marker genes mentioned previously (Figure 5). These genes are PCSK1, TGFBI, PMM1, CLDN7, NPPB, and CRYAB in SMC and PCSK1, C1S, CALCOCO2, SAC3D1, MAOA, RGCC, and ADH1B in SMC-calc. The TGFBI gene was the only gene selected by the random forest, limma and STAMP programs for the SMC group.
Protein–protein interaction (PPI) network analysis of the 12 candidate genes, using STRING and MCL clustering, revealed six functional clusters separated by dashed lines; the thickness of the lines reflects the confidence level of the interaction (Figure 6). PCSK1, RGCC, SAC3D1, TGFBI, and C1S genes did not form highly connected modules in the global network. Therefore, individual analyses were performed on these genes as peripheral modules (Figure 6). The dominant ADH1 and MAOA cluster is predominantly composed of members of the alcohol and aldehyde dehydrogenase family and other oxidoreductases, characterized as being involved in drug metabolism. In addition, the CLDN7 cluster is associated with the positive regulation of blood–brain barrier permeability, the AOC1 cluster with amine metabolism and local vascular regulation, the NPPB cluster with natriuretic peptide as a vasoactive factor, the CALCOCO2 cluster with autophagy and immunity, and the CRYAB cluster with eye lens protein function. Peripheral modules identified by MCL included clusters involved in peptide hormone regulation (centered on PCSK1), phosphorylation of Early Mitotic Inhibitor 1 (Emi1) (RGCC cluster), extracellular matrix remodeling (TGFBI cluster), transcription export complex 2 related to ribosomal and RNA processing (SAC3D1 cluster), and complement-mediated immune response (C1S cluster). Overall, this network analysis shows that these genes participate in coordinated yet diverse biological modules, encompassing processes such as vascular remodeling, metabolic regulation, cellular homeostasis, and immune defense. This suggests their functional connectivity in the mechanisms of vascular disease and their potential value as candidate genes for translational biomarkers. (Line 269-297)
(Line 336-376) Among the 12 identified markers, several have been previously implicated in SMC dedifferentiation and vascular pathology. PCSK1, a proprotein convertase, has been identified as a promising therapeutic target for cardiovascular diseases [43]. TGFBI is an extracellular matrix protein that increases the proliferation of airway SMCs [44], and it has been associated with stimulating SMC migration [45]. PMM1 is an enzyme involved in the synthesis of GDP-mannose, and plasma mannose levels are related to incident type 2 diabetes and cardiovascular disease [46]. Furthermore, CLDN7, a tight junction protein, suppresses breast cancer invasion and metastasis through the repression of the smooth muscle actin program [47]. NPPB, a hormone, may play a protective role in vascular disease by counteracting the effects of vasoconstrictors [48]. CRYAB is a small heat shock protein that acts as a molecular chaperone, and it is upregulated in injury-induced vascular SMC proliferation [49]. Additionally, complement activation contributes to vascular damage in atherosclerosis, and C1S may be involved in promoting inflammation and SMC dysfunction in atherosclerotic lesions [50]. CALCOCO2, also known as NDP52, has been studied primarily in the context of autophagy and innate immunity [51]. However, its role in SMCs is not well defined. SAC3D1 is involved in the cell cycle, centrosome duplication and spindle formation [52]. Although its role in SMCs is uncertain, it has been used as a prognostic marker in cancer [53]. MAOA is an enzyme that breaks down neurotransmitters such as serotonin, dopamine, and norepinephrine. MAOA affects the contractile activity of smooth muscle gastric tissues through 5-hydroxytryptamine [54]. RGCC may promote injury-induced vascular neointima formation by mediating SMC proliferation and migration, thus facilitating atherosclerosis development [55]. The role of ADH1B in SMCs is not well understood, although it is downregulated in airway smooth muscle by epidermal growth factor and interleukin-1β [56].
Among the 12 markers identified in this study, many are already known to play important roles in smooth muscle cell biology and vascular disease, which shows their value for translation. Our PPI network analysis further showed that these genes are grouped into specific functional clusters, not randomly scattered. These modules include vascular remodeling and fibrosis (TGFBI), metabolic and redox functions (ADH1/MAOA cluster), hormone-related pathways (PMM1 and NPPB), cell junction structure (CLDN7), immune and autophagy functions (C1S and CALCOCO2), and stress response (CRYAB). For example, the ADH1 and MAOA cluster is related to metabolism and vascular diseases, and may be useful as a biomarker or treatment target. The CLDN7, NPPB, CALCOCO2, and CRYAB clusters have functions in maintaining cell stability, regulating blood vessel contraction, protecting against immune stress, and helping the cell withstand damage. These network results confirm that the genes work together in connected biological processes that are key for smooth muscle cell changes and vascular disease. Therefore, using PPI network analysis helps us understand how these genes can be applied, particularly for developing diagnostic tests, assessing risk, or designing individualized treatments for vascular disease [44-51].

Round 2
Reviewer 1 Report
Comments and Suggestions for Authors
Authors have addressed all the major comments and updated the paper accordingly.
Reviewer 2 Report
Comments and Suggestions for Authors
Thank you for making the changes to the manuscript.